# Systematic review of social determinants of childhood immunisation in low- and middle-income countries and equity impact analysis of childhood vaccination coverage in Nigeria

**Sarah V. Williams**[1]*, **Tanimola Akande**[2], **Kaja Abbas**[3,4]

**1** Royal Free London, NHS Foundation Trust, London, United Kingdom, **2** Department of Epidemiology & Community Health, University of Ilorin, Ilorin, Nigeria, **3** Department of Infectious Disease Epidemiology and Dynamics, London School of Hygiene & Tropical Medicine, London, United Kingdom, **4** School of Tropical Medicine and Global Health, Nagasaki University, Nagasaki, Japan

* sarah.williams122@nhs.net

## Abstract

### Background

Nigeria has a high proportion of the world's underimmunised children. We estimated the inequities in childhood immunisation coverage associated with socioeconomic, geographic, maternal, child, and healthcare characteristics among children aged 12–23 months in Nigeria using a social determinants of health perspective.

### Methods

We conducted a systematic review to identify the social determinants of childhood immunisation associated with inequities in vaccination coverage among low- and middle-income countries. Using the 2018 Nigeria Demographic and Health Survey (DHS), we conducted multiple logistic regression to estimate the association between basic childhood vaccination coverage (1-dose BCG, 3-dose DTP-HepB-Hib (diphtheria, tetanus, pertussis, hepatitis B and Haemophilus influenzae type B), 3-dose polio, and 1-dose measles) and socioeconomic, geographic, maternal, child, and healthcare characteristics in Nigeria.

### Results

From the systematic review, we identified the key determinants of immunisation to be household wealth, religion, and ethnicity for socioeconomic characteristics; region and place of residence for geographic characteristics; maternal age at birth, maternal education, and household head status for maternal characteristics; sex of child and birth order for child characteristics; and antenatal care and birth setting for healthcare characteristics. Based of the 2018 Nigeria DHS analysis of 6,059 children aged 12–23 months, we estimated that basic vaccination coverage was 31% (95% CI: 29–33) among children aged 12–23 months, whilst 19% (95% CI:18–21) of them were zero-dose children who had received none of the basic vaccines. After controlling for background characteristics, there was a significant

**Data Availability Statement:** The 2018 Nigeria DHS data set is publicly available and accessible upon registration for legitimate research purposes

on the DHS website at https://dhsprogram.com/methodology/survey/survey-display-528.cfm. To download DHS datasets, researchers must register as a DHS data user at https://dhsprogram.com/data/new-user-registration.cfm.

**Funding:** KA is supported by Save the Children, Vaccine Impact Modelling Consortium (INV-034281), and the Japan Agency for Medical Research and Development (JP223fa627004). The funders had no role in study design, data collection and analysis, decision to publish, or preparation of the manuscript. The authors declare that they have no known competing financial interests or personal relationships that could have appeared to influence the work reported in this paper.

**Competing interests:** The authors have declared that no competing interests exist.

**Abbreviations:** AOR, Adjusted odds ratio; BCG, Bacille Calmette-Guérin vaccine; DHS, Demographic Health Survey; DTP, Diphtheria, tetanus and pertussis containing vaccine; DTP-HepB-Hib, Diphtheria, tetanus, pertussis, hepatitis B and *Haemophilus influenzae* type B; ECI, Erreygers concentration indices; EPI, Expanded Programme of Immunization; Gavi, Global Alliance for Vaccines and Immunisations; LGA, Local Government Area; LMIC, Low- and Middle-Income Countries; PRISMA, Preferred Reporting Items for Systematic Reviews and Meta-Analyses; SDG, Sustainable Development Goals; UNICEF, United Nations Children's Fund; WHO, World Health Organisation; WUENIC, WHO and UNICEF estimates of national immunization coverage.

increase in the odds of basic vaccination by household wealth (AOR: 3.21 (2.06, 5.00), p < 0.001) for the wealthiest quintile compared to the poorest quintile, antenatal care of four or more antenatal care visits compared to no antenatal care (AOR: 2.87 (2.21, 3.72), p < 0.001), delivery in a health facility compared to home births (AOR 1.32 (1.08, 1.61), p = 0.006), relatively older maternal age of 35–49 years compared to 15–19 years (AOR: 2.25 (1.46, 3.49), p < 0.001), and maternal education of secondary or higher education compared to no formal education (AOR: 1.79 (1.39, 2.31), p < 0.001). Children of Fulani ethnicity in comparison to children of Igbo ethnicity had lower odds of receiving basic vaccinations (AOR: 0.51 (0.26, 0.97), p = 0.039).

## Conclusions

Basic vaccination coverage is below target levels for all groups. Children from the poorest households, of Fulani ethnicity, who were born in home settings, and with young mothers with no formal education nor antenatal care, were associated with lower odds of basic vaccination in Nigeria. We recommend a proportionate universalism approach for addressing the immunisation barriers in the National Programme on Immunization of Nigeria.

## Introduction

Nigeria is the most populous country in Africa with around 202 million people in 2020 and its population is predicted to double by 2050 [1]. It is a multi-ethnic country with 36 autonomous states and the Federal Capital Territory. Around 83 million people (40% of total population) live below the poverty line while an additional 53 million people (25% of total population) are vulnerable to falling below the poverty line [2]. Economic growth has been slow with challenges including ongoing conflict in parts of the country, inconsistent regulatory environment, poor power supply and infrastructure [2].

Vaccination is a highly cost-effective public health intervention and beyond the direct benefits to population health, vaccines provide additional economic and social benefits to individuals and society [3, 4]. Infectious diseases remain a leading cause of death among under-5-year-old children, and an additional 1.5 million deaths could be avoided every year with improvements in global vaccination coverage [5, 6]. Model-based estimates, not including COVID-19, project 51 million deaths to be prevented by vaccination during 2021–2030 [7]. There have been substantial improvements in vaccine introductions and vaccination coverage in low- and middle-income countries since the inception of Gavi, the Vaccine Alliance in 2000 [8]. However, the prevalence of zero-dose children, that is children aged 12–23 months who had not received any of the routine childhood vaccines, was 7.7% in low- and middle-income countries during 2010–2019 [9]. The importance of improving vaccination coverage was recognised in the Sustainable Development Goals (SDGs), with immunisation contributing to 14 of the 17 SDGs and includes reduction on poverty, hunger and improving social equity [10]. Vaccination coverage and equity are a strategic goal of the global Immunisation Agenda 2030, with the aim to reach equitable coverage at national and district levels by addressing immunisation barriers posed by location, age, socioeconomic status, and gender [11].

The Expanded Programme on Immunisation (EPI) was established by the World Health Organization (WHO) in 1974 to improve vaccination services globally [12], and Nigeria began nationwide implementation of EPI in 1979 which was later changed to the National

Programme on Immunization [13]. Although the vaccines in the routine immunisation programme (S1 Table) for under 5-year-old children are available with no out-of-pocket charges [14], Nigeria has the most under-immunised children in the world with 4.5 million in 2018 [15]. The immunisation system challenges in Nigeria include weak institutions, service delivery, funding, infrastructure, poor coordination between the National Programme on Immunization and non-governmental organisations delivering vaccination services, and a lack of political commitment in some regions, with further challenges to immunisation services caused by the COVID-19 pandemic [14, 16–18]. There are fewer adequately skilled healthcare personnel in rural areas and northern states, and poor retention and frequent transfers of workers. Security is also an issue, with attacks on healthcare workers in recent years. Attitudes of communities and caregivers are important too, with a lack of knowledge about vaccination and mistrust of services hindering vaccination uptake [16, 19].

In the context of wider immunisation system challenges in Nigeria, we focused on factors associated with inequities in basic vaccination coverage through the social determinants of health model. This model framework has been explicitly linked to health equity by the WHO Commission on Social Determinants of Health [20] and considers the social, cultural, political, economic, commercial and environmental factors that shape the conditions in which people are born, grow, live, work and age, and these factors are determined by wealth, power and resources. In this study, we use the social determinants of health model framework, which encompasses the individual, parental, household, environment, and national policy levels that influence inequities in basic vaccination coverage among children in Nigeria (see Fig 1). We refer to vaccine inequity as unfair and avoidable or remediable differences in health among population groups defined socially, economically, demographically, or geographically [21]. This is related to but distinct from health inequality, which indicates the status of imbalances or differences in health among population groups without any moral judgement on whether the imbalances or differences are fair or not [22, 23].

Our aim is to analyse the 2018 Nigeria Demographic and Health Survey (DHS) and estimate the inequities in basic vaccination coverage (1-dose BCG (Bacille Calmette-Guérin), 3-dose DTP-HepB-Hib (diphtheria, tetanus, pertussis (DTP), hepatitis B (HepB) and *Haemophilus Influenzae* type b (Hib)), 3-dose polio, and 1-dose measles vaccines) associated with socioeconomic, geographic, maternal, child, and healthcare characteristics among children aged 12–23 months in Nigeria. We conducted disaggregated equity impact analysis to reveal the inequities in basic vaccination coverage that are hidden at the aggregated national level, and understand the facilitators and barriers to vaccination through the social determinants of health framework.

## Methods

### Study design of demographic and health survey

We analysed the 2018 Nigeria DHS which was conducted between August to December 2018 [24]. The DHS are nationally representative household surveys focusing on population, health, and nutrition in LMICs [25]. The DHS sample is a two-stage stratified cluster sample with sampling weights applied to ensure that results are representative. There are four questionnaires: household questionnaire, woman's questionnaire, man's questionnaire, and biomarker questionnaire. The country is divided into clusters with 30 households selected from each cluster. The woman's questionnaire was asked to women aged 15–49 years and provides the data for our study. All women aged 15–49 years in the sampled households were included and the survey was successfully conducted in 1,389 clusters after 11 clusters were dropped following deteriorating security in those areas during data collection. In addition, in the state of Borno,

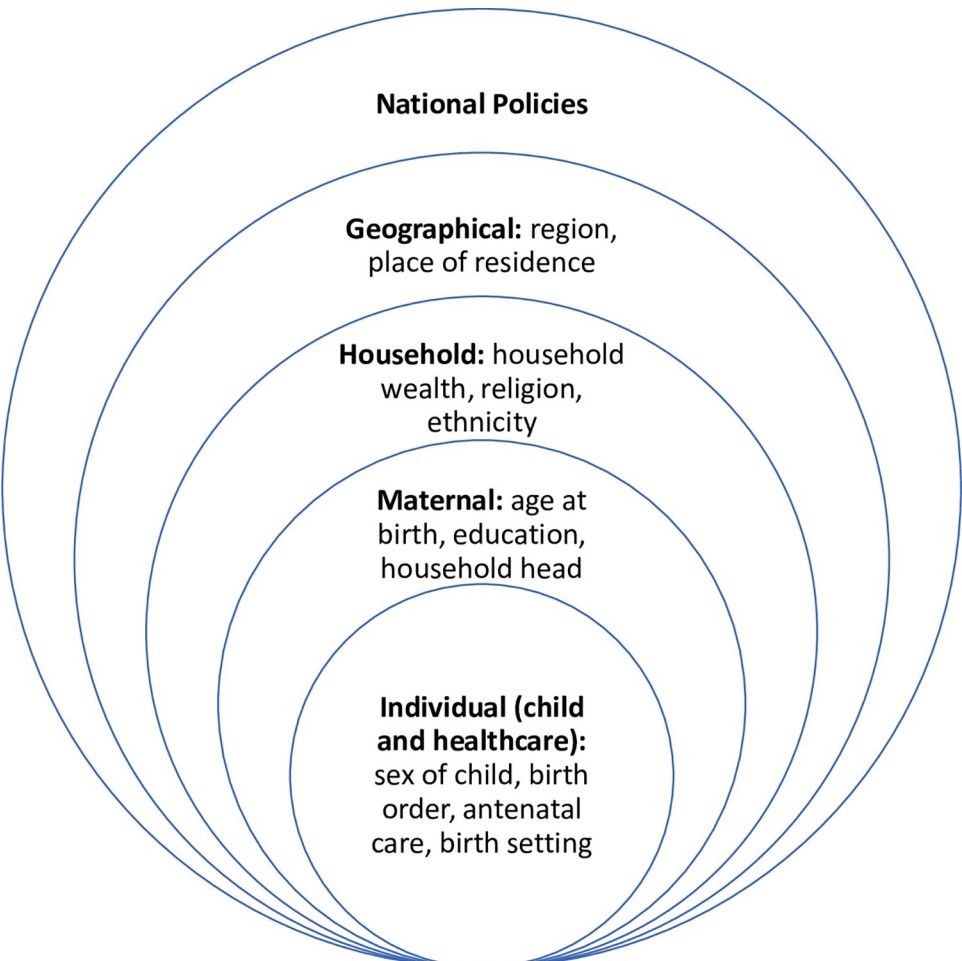

**Fig 1. Social determinants of childhood immunisation.** Social determinants of health model framework encompassing the individual, parental, household, environment, and national policy levels and influencing inequities in basic vaccination coverage among children in Nigeria.

11 of the 27 Local Government Areas (LGAs) in the state were dropped due to insecurity. Clusters selected from these dropped LGAs were replaced with clusters from the remaining LGAs and so may not be representative of the entire state [24]. The study population for our analysis was women aged 15–49 years with a child aged 12 to 23 months old. Of the 41,821 women interviewed, 33,924 women had a child aged 59 months or younger and immunisation data was collected for 6,059 living children aged between 12–23 months. We applied sampling weights to the survey dataset to adjust for disproportionate sampling and non-response, thereby ensuring that the sample was representative of the population.

## Characteristics selection and systematic review

The WHO had analysed inequality in immunisation using the following characteristics determined from a literature review: child (gender, birth order), maternal (age at birth, education, ethnicity or caste), household (sex of household head, household economic status), geographic (rural or urban place of residence, subnational region) [26]. Using the WHO inequality work as a starting point, we conducted a systematic review to select the pertinent DHS variables for our equity analysis. We conducted a systematic review to analyse qualitative and quantitative

peer-reviewed literature of studies focused on inequities in childhood immunisation in low- and middle-income countries (LMICs) to identify the social determinants of childhood immunisation. S1 Fig illustrates the process flow diagram of identification, screening, eligibility, and inclusion of articles for the systematic review, using the PRISMA (Preferred Reporting Items for Systematic Reviews and Meta-Analyses) framework (S1 Checklist) [27]. We conducted our search using MEDLINE and additional studies were identified through hand-searches of reference lists for articles written in the English language, published between 01/01/2010 to 04/10/2021, for which the full text was available, and contained the following terms in English or American spellings: (Vaccination coverage or Immunisation coverage) AND (Determinant, characteristic, predictor) AND (Equity, equality, disparity, inequality, inequity). We did not search prior to 2010 as papers and the subsequently identified social determinants may be less relevant to the current context.

## Vaccination coverage

In the 2018 Nigeria DHS, the information of whether a child has received a vaccination is gathered from the child's vaccination card. If that is not available or if a vaccine has not been recorded, then the mother is asked which vaccines have been given to her child [24]. Our primary outcome and dependent variable of interest was basic vaccination coverage, that is the proportion of children receiving 1-dose BCG, 3-dose DTP-HepB-Hib, 3-dose polio, and 1-dose measles vaccines. DHS gives the vaccination status of "received" and "not received" for individual vaccines. Binary variables of "received all three doses" and "not received all three doses" were generated for the three dose vaccines. The basic vaccination variable was generated as a binary variable of "received all basic vaccinations" and "not received all basic vaccinations" by combining the variables for 1-dose BCG, 3-dose DTP-HepB-Hib, 3-dose polio, and 1-dose measles vaccines.

## Equity analysis

We conducted simple logistic regression to estimate crude odds ratios and assess basic vaccination coverage disaggregated by socioeconomic, geographic, maternal, child, and healthcare characteristics as measured by the DHS and identified through the systematic review and the WHO inequality report on immunisation [26]. Due to the survey design, a p-value for all categories within a characteristic was determined using the Adjusted Wald's test through the "test" function in Stata [28]. These p-values were used to assess the association between basic vaccination coverage and the characteristics [29] and if a p-value was <0.05 then the characteristic was included in the model. We checked for collinearity between variables and avoided multicollinearity in the development of a parsimonious model [29] using the "regress" in Stata and calculation of the variance inflation factor. However, no variables required removal from the model.

Interaction was tested using the "contrast" function in Stata. This gives F statistics which are adjusted for the survey design, with the p-value demonstrating the statistical significance of the interaction. As in the WHO report [26, 29], interactions were tested between mother's education and household wealth, mother's age and household wealth, mother's age and education, and place of residence and household wealth. We conducted multivariable logistic regression to estimate adjusted odds ratios (AORs) for socioeconomic, geographic, maternal, child, and healthcare characteristics associated with basic vaccination coverage.

We analysed inequity further by estimating the Erreygers concentration indices for maternal education, antenatal care, and household wealth to assess if basic vaccination coverage and vaccination card usage had progressive, regressive, or equal distribution based on each of these

characteristics. The concentration index value shows how much of a health measure is concentrated in an advantaged or disadvantaged group. Values range from +1 to -1, with a value of zero meaning there is no inequity and positive values indicating that a health measure is concentrated in the more advantaged groups [30].

### Reproducibility of analysis

We conducted the survey analysis using the Stata statistical software [28], and visualisations were generated using the R statistical software [31].

## Results

In the systematic review, we identified 160 publications, screened the title and abstract, assessed full articles for eligibility, and included 49 publications in our systematic review (see Table 1). In addition to the characteristics analysed by the WHO inequality report on immunisation [26], we identified that antenatal care and birth setting had evidence of association with vaccination coverage. We streamlined the social determinants of childhood immunisation to: household wealth, religion, and ethnicity for socioeconomic characteristics; region and place of residence for geographic characteristics; maternal age at birth, maternal education, and maternal household head status for maternal characteristics; sex of child and birth order for child characteristics; and antenatal care and birth setting for healthcare characteristics.

### Vaccination coverage

Among the 6,059 children aged 12–23 months in the 2018 Nigeria DHS following the application of sample weights, 2,100 (35%) of them lived in urban areas. The coverage for the individual vaccines of BCG, DTP-HepB-Hib, polio, and measles was higher in urban areas compared to rural areas (Fig 2 and S2 Table). At the national level, the mean basic vaccination coverage was 31% (95% CI: 29–33) with coverage of single vaccinations ranging from 48% (46–50%) for the third dose of the polio vaccine to 73% (71–75%) for the first dose of the polio vaccine.

Vaccination cards were available for 49% (46–51%) of children, among which 57% (55–60%) had received all basic vaccinations. Among the 51% (49–54%) of children without vaccination cards, only 6.4% (5.3–7.7%) of them had received all basic vaccinations. Almost one fifth of children (19% (18–21%)) had not received any of the basic vaccinations and in rural areas almost a quarter had received none. For polio vaccinations, 26% (24–28%) of children had received none of the three doses and for DTP vaccinations, 35% (33–37%) of children had received none. There was a higher proportion of zero-dose children in rural areas compared to urban areas.

### Equity analysis

Fig 3 shows the basic vaccination coverage among children aged 12–23 months in Nigeria disaggregated by socioeconomic (household wealth, religion, ethnicity), geographic (region, place of residence), maternal (maternal age at birth, maternal education, maternal household head status), child (sex of child, birth order), and healthcare (antenatal care, birth setting) characteristics. For socioeconomic characteristics, children living in wealthier households had higher basic vaccination coverage ranging from 17% (15–20%) to 49% (45–53%) from the poorest to the richest wealth quintiles. For religion, basic vaccination coverage was higher among children of the Catholic faith at 49% (43–54%) while most children (58%) were of Islamic faith with relatively lower coverage of 23% (21–25%). Regarding ethnicity, basic vaccination coverage ranged from 12% (9–16%) to 56% (51–61%) among children of Fulani and Igbo ethnicities

**Table 1. Systematic review of social determinants of childhood immunisation.**

| Location | Key inferences | Source | |
|---|---|---|---|
| Afghanistan | Approximately 60% of children aged 1–4 years were under vaccinated or not vaccinated, with large disparities among provinces and higher coverage in urban areas. Increased odds of full immunisation were associated with giving birth in government institution, having a higher number of antenatal care visits, and visiting a health facility in the past 12months. Maternal involvement in household decision making was also significantly associated with vaccination status. | Shenton et al., 2018 | [32] |
| Angola | In Bom Jesus in Angola, 37% of under 5-year-old children had completed the vaccination schedule, but coverage was higher for under 1-year-old children. Coverage was higher in rural areas. The percentage of children completing the vaccination schedule varied according to child age, mother's education, family size, ownership of household appliances, and means of disposal of domestic waste. | Oliveira et al., 2014 | [33] |
| Bangladesh | There was no significant disparity of vaccination coverage by ethnicity. There was significant variation of vaccination coverage by the child's gender (lower for females), household ownership of mobile phones and by household geographical location. | Rahman et al., 2018 | [34] |
| Bangladesh | Maternal age, education and wealth have a significant effect on coverage of most vaccines. The number of microfinancing organisations in the community also increased the odds of children being vaccinated, whilst poor accessibility or being far from a health centre, specifically an Upazilla Health Complex, decreased the odds of a child being vaccinated. There are differences between regions, but even in regions with relatively higher coverage, some communities had poor coverage. | Vyas et al., 2019 | [35] |
| Cameroon | Analysis of five rounds of DHS surveys from 1991 to 2001 showed that children in the latter rounds were more likely to be fully immunised. The likelihood of children being fully immunised was affected by having a vaccination card, birth order, maternal age and maternal education. | Nda'chi Deffo et al., 2020 | [36] |
| Democratic Republic of the Congo | In urban areas, children of educated mothers had higher measles vaccination coverage in comparison to children of mothers with no formal education. In rural areas, children of wealthier households had higher measles vaccination coverage. Vaccination card usage rates were higher in urban areas than rural areas. | Ashbaugh et al., 2018 | [37] |
| Ethiopia | Around 25% of children aged 12–23 months were fully vaccinated and coverage varied with urbanicity. Predictors of full coverage were sources of information from the vaccination card, received postnatal check-up within two months after birth, region, women's awareness of community conversation program, and wealth index. | Lakew et al., 2015 | [38] |
| Ethiopia | Children from poorer households, rural regions of Afar and Somali, no maternal education, and female-headed households had lower full vaccination coverage. | Geweniger et al., 2020 | [39] |
| Ghana | Vaccination coverage at the end of the first year of life was high except for polio vaccine given at birth. However, many vaccines were given late and there was substantial health inequity across socioeconomic indicators for timeliness of vaccination. | Gram et al., 2014 | [40] |
| India | Slightly more than half of children aged 12–36 months were fully vaccinated in 2008. The analysis adjusted for state of residence, age, gender, household wealth, and maternal education, and other predictors of vaccination were religion, caste, place of delivery, number of antenatal care visits, and maternal tetanus vaccination. Children in urban areas had higher odds of being unvaccinated or under-vaccinated than those in rural areas. | Shrivastwa et al., 2015 | [41] |
| India | Vaccine coverage in an urban poor area of south east Delhi was lower than estimates from other studies for overall regional urban coverage, indicating a discrepancy in coverage between the urban poor and urban non-poor. Coverage was determined by gender of child, religion, maternal literacy, household's socioeconomic position, being born in a facility and being born outside of Delhi and having a birth certificate. | Devasenapathy et al., 2016 | [42] |
| India | There were regional disparities in vaccination coverage with a north south divide. All doses of vaccination coverage were significantly associated with postnatal care, institutional births, neonatal tetanus protection of the last birth, women's education, and health insurance coverage | Khan et al., 2018 | [43] |
| India | Approximately 50% of children aged 12–23 months were fully vaccinated in a pooled dataset of three time periods (1998–99, 2002–04 and 2007–08), and children recorded in the 2007–08 dataset were more likely to be vaccinated. Vaccination was inversely associated with female gender, Muslim religion, lower caste, urban residence, and maternal characteristics (lower educational attainment, non-institutional delivery, fewer antenatal care visits and non-receipt of maternal tetanus vaccination). The most common reason for non-vaccination was that mothers were unaware of the need. | Francis et al., 2018 | [44] |
| India | Immunisation was highly associated with maternal education and household wealth. Inequality in immunisation was highest in lower socioeconomic groups among children of mothers with no formal education. | Kannankeril Joseph VJ et al., 2021 | [45] |
| India | Full immunisation was associated with living in urban areas and richer household wealth, and was highest in Manipur and lowest in Nagaland. | Srivastava et al., 2021 | [46] |

*(Continued)*

**Table 1.** (Continued)

| Location | Key inferences | Source | |
|---|---|---|---|
| Indonesia | Immunisation was significantly associated with birth order, maternal age, maternal education, paternal occupation, antenatal care, and living in certain regions. Being resident in communities with a higher proportion of public health centres were more likely to be fully immunised. | Siramaneerat et al., 2021 | [47] |
| Kenya | Measles vaccination coverage differed according to household wealth, parents' education, skilled antenatal care visits, birth order and father's occupation. Rural residence reduced inequality in measles vaccination, which may reflect efforts to provide vaccination sites in remote and rural areas of the country. | Van Malderen et al., 2013 | [48] |
| Kenya | Birth setting, ethnicity, and wealth index were significant predictors of vaccination coverage. Somalis had greater odds of being under or non-vaccinated than the Kikuyu ethnic group, and wealth and birth setting were associated with immunisation status for both Somalis and non-Somalis. | Masters et al., 2019 | [49] |
| Kenya | Children of mothers with no education, born in home settings, in regions with limited health infrastructure, living in poorer households, and of higher birth order are associated with lower rates of full immunisation. | Allan et al., 2021 | [50] |
| Kenya | Immunisation coverage and timeliness of immunisation improved during 2003–2017. In two urban informal settlements, the hazard for being fully immunised varied by household wealth and ethnicity. | Mutua et al., 2020 | [51] |
| Laos | Community demand for immunisation was lowered by health system barriers of multiple providers, inconsistent record keeping, and inadequate health information system. Lack of understanding of the value of vaccination and immunisation services was demonstrated at the individual and household levels. | Sychareun et al., 2019 | [52] |
| Malawi | There were regional differences in vaccination coverage and regions with higher coverage had a high percentage of deliveries attended by a health professional. Coverage was higher for women who gave birth at a hospital or maternity clinic or had a midwife or nurse at birth. Coverage was significantly correlated with some socio-demographic characteristics: child's age, illiteracy, income, water, and sanitary conditions. | Abebe et al., 2012 | [53] |
| Malawi | In children aged 12–23 months, individual factors had a stronger effect than community factors on vaccination coverage. Mother's education, frequency of antenatal care visits, use of immunisation cards, household wealth, and geographical region were the most significant factors associated with vaccination coverage. | Ntenda et al., 2017 | [54] |
| Mozambique | Analysis of data from the 2015 Immunization, AIDS and Malaria Indicators Survey (IMASIDA), indicated that living in the Southern region was a protective factor for full immunisation whilst being in the poorest quintile increased the risk of a child not being fully immunised. | Daca et al., 2020 | [55] |
| Nepal | A quarter of children were not fully immunised. Full immunisation was associated with having an immunisation card, delivery in a public or private institution, maternal education and maternal employment. | Patel et al., 2021 | [56] |
| Nigeria | Socio-economic characteristics explained the disparities in full vaccination and recommended community-level interventions focused on improving full vaccination coverage. | Antai, 2009 | [57] |
| Nigeria | Women who were sole providers of household earnings were associated with higher likelihood of fully immunising their children, while women who lack decision-making autonomy were associated with lower likelihood of fully immunising their children. | Antai, 2012 | [58] |
| Nigeria | Routine immunisation data shows lower coverage in states of northern Nigeria compared with states of southern Nigeria. When comparing determinants of supply side access in two northern and two southern states, regional supply-side disparities were not apparent. However, there was a general sub-optimal supply of services and residents in the northern states were more likely to live within 5km of an immunisation service. This supports socio-cultural explanations of disparities in immunisation coverage. | Eboreime et al., 2015 | [59] |
| Nigeria | Fully immunised children are concentrated amongst the rich and partially immunised children are concentrated amongst the poor, no significant difference between socioeconomic status for unimmunised. Concentration of fully immunised children among the rich were determined by mother's literacy, living in the rural area, socioeconomic status and geopolitical location. Significant concentration of partially immunised children among the poor were determined by the same factors. | Ataguba et al., 2016 | [60] |
| Nigeria | In a cross-sectional study in south-eastern Nigeria of under 5-year-old children, vaccination coverage was poor and below national targets. There were differences in coverage between wealth quartiles with the wealthiest having higher coverage than the poorest. Coverage was highest for children in the first year of life and decreased with age. | Uzochukwu et al., 2017 | [61] |
| Nigeria | Children being unvaccinated against polio was significantly associated with household wealth index, maternal educational level, maternal employment, geopolitical and neighbourhood illiteracy level. | Uthman et al., 2017 | [62] |

(*Continued*)

**Table 1.** (Continued)

| Location | Key inferences | Source | |
|---|---|---|---|
| Nigeria | Coverage varied by place of residence (rural, urban formal, and urban slum) and was significantly associated with place of delivery, antenatal care, maternal education, maternal age at childbirth, religion, place of residence, media exposure and distance to a health facility. Population attributable risk analysis demonstrated the biggest increase in coverage for maternal education and biggest reduction for non-attendance at antenatal care. | Obanewa et al., 2020 | [63] |
| Pakistan | Vaccination coverage of children aged 12–23 months was strongly associated with maternal education, paternal education, extended family structure and family wealth. Coverage differed by ethnicity, with Bengali children having the lowest coverage. | Siddiqui et al., 2014 | [64] |
| East African countries | Most children in each country had received one of the recommended vaccinations, with regional variation within countries. There were no consistent predictors across countries of complete vaccination status. However, being delivered in a public institution rather than at home was associated with increased odds, except in Burundi. Whether a child had received a check-up within two months of birth was also associated with complete vaccination status in Burundi, Kenya, and Uganda. | Canavan et al., 2014 | [65] |
| West Africa | Vaccination coverage increased between 2000 and 2017. Inequalities in coverage were mainly related to poverty, maternal education and living in certain regions. | Wariri et al., 2019 | [66] |
| Sub-Saharan Africa | Poor households were more likely to have missed opportunities for vaccination. In some countries, literacy, maternal education, media access and the number of under 5-year-old children contributed to missed opportunities for vaccination. | Ndwandwe et al., 2018 | [67] |
| Sub-Saharan Africa | Education inequality in missed opportunities for vaccination was mainly explained by differential effects such as neighbourhood socioeconomic status, presence of under 5-year-old children, media access and household wealth index. | Sambala et al., 2018 | [68] |
| Sub-Saharan Africa | Vaccination coverage was lower in rural areas and household wealth, birth order and distance to health facilities were important contributors to the rural-urban gap. | Ameyaw et al., 2021 | [69] |
| Conflict-impacted countries | In conflict affected countries, low vaccination coverage and outbreaks of vaccine preventable diseases are a concern. Many outbreaks were in conflict-affected areas or in displaced populations. In 14 of the 16 conflict affected countries, DTP3 coverage was below the global average of 85% in 2014. | Grundy et al., 2019 | [70] |
| Gavi-supported countries | Following a systematic analysis of inequalities in vaccination coverage in 45 Gavi supported countries, recommendations were made for Gavi equity monitoring. In addition to the wealth index, other measures of vulnerability should be monitored: maternal education, place of residence, child sex and the multidimensional poverty index. Both absolute and relative measures should be tracked. | Arsenault et al., 2017 | [71] |
| Gavi-supported countries | DHS data from 45 Gavi supported countries demonstrated that vaccination coverage and inequalities varied between countries. There were wealth, education, and multidimensional poverty index (MPI) inequalities in vaccination coverage. Country level predictors of vaccination coverage included political stability, government expenditure on health, government effectiveness and control of corruption, greater land area, linguistic fractionalisation, and gender inequality. There were similar associations across the three indicators of inequality (wealth, maternal education and multidimensional poverty) for country level predictors of inequality. | Arsenault et al., 2017 | [72] |
| Low- and middle-income countries | By analysing data from 241 nationally representative household surveys in 96 countries, being unvaccinated was strongly associated with education of the caregiver, education of caregiver's partner, caregiver's tetanus toxoid status, wealth index and type of family member participation in decision-making when the child is ill. Tetanus toxoid status was the strongest predictor of unvaccinated status. | Bosch-Capblanch et al., 2012 | [73] |
| Low- and middle-income countries | Twenty-five studies were reviewed to assess the relevance of gender inequality on the structural, health system, community, and individual levels. Vaccination programmes often target mothers and can therefore reinforce a community's gender and social dynamics, with women facing barriers at every level to accessing vaccinations: access to education, income, as well as autonomous decision-making about time and resource allocation were evident barriers. | Merten et al., 2015 | [74] |
| Low- and middle-income countries | The greatest disparities were among children born to women with no formal education compared to children born to women with secondary or higher education. Coverage was lower in rural areas compared to urban areas and in the lowest wealth quintile compared to the richest quintile. No gender differences were observed. | Hinman et al., 2015 | [75] |
| Low- and middle-income countries | Rural-urban migrant children were less likely to be immunised than urban non-migrants and the general population. Coverage estimates were lower for most vaccines for rural-urban migrants than for the general population. | Awoh et al., 2016 | [76] |
| Low- and middle-income countries | Vaccination coverage varied between and within countries, and coverage was lowest in children from poorer households with pro-rich inequality in most countries. | Hosseinpoor et al., 2016 | [77] |

(*Continued*)

**Table 1.** (Continued)

| Location | Key inferences | Source | |
|---|---|---|---|
| Low- and middle-income countries | In most countries, vaccination coverage was pro-rich, and coverage was higher in urban areas than rural areas. Antenatal care was concentrated among wealthier mothers and was significantly associated with the concentration of vaccination coverage among wealthier children. | Hajizadeh et al., 2018 | [78] |
| Low- and middle-income countries | The WHO analysed inequality in immunisation using the following characteristics determined from a literature review: child (gender, birth order), maternal (age at birth, education, ethnicity or caste), household (sex of household head, household economic status), geographic (place of residence—rural or urban, subnational region). The analysis focused on ten countries that account for more than 70% of children globally who did not receive basic vaccination. | WHO, 2018 | [26] |
| Low- and middle-income countries | DHS data was examined for socioeconomic inequalities in the completion rates of vaccines in the Gambia, the Kyrgyz Republic and Namibia. Higher completion of vaccination was observed in rural areas. Completion rate for BCG, DTP3, polio (3-doses), and measles vaccines had a pro-poor distribution with children of lower socioeconomic status more likely to be immunised than children from higher socioeconomic status. In the Gambia and Namibia, the difference in completion rates for childhood immunisation between rural and urban areas was the primary contributor for the concentration of child vaccination among the poor. In the Kyrgyz Republic, household wealth was a key determinant for child vaccination. Birth order, maternal age, maternal education, and skilled birth attendance were also factors in some of the countries. | Hajizadeh,. 2019 | [79] |

Systematic review of studies focused on inequities in childhood immunisation in low- and middle-income countries.

respectively. For geographic characteristics, 65% of children lived in rural areas but basic vaccination coverage was higher for children in urban areas at 44% (41–47%) in comparison to 23% (21–25%) in rural areas. At the regional level, basic vaccination coverage among children ranged from 20% (17–23%) in the North West region to 57% (51–62%) in the South East region (Fig 4). For maternal characteristics, children of mothers aged 35–49 years had higher basic vaccination coverage at 34% (31–39%) in comparison to 16% (12–21%) for children of younger mothers aged 15–19 years. Basic vaccination coverage increased with higher levels of maternal education, with basic vaccination coverage among children of mothers with no

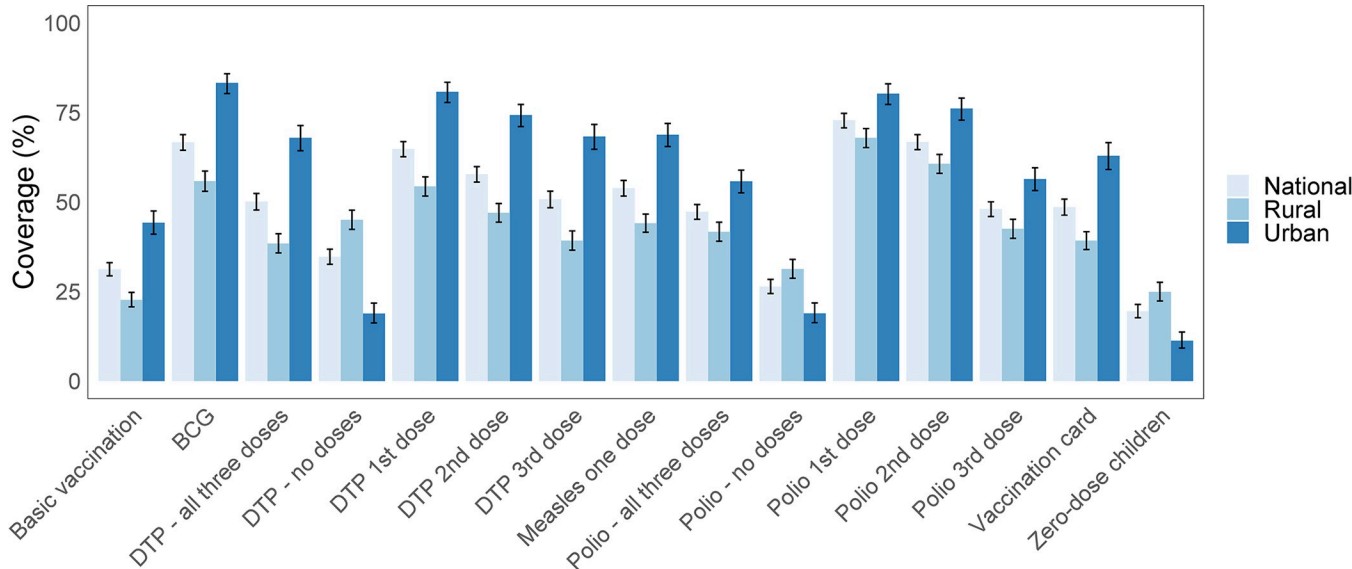

**Fig 2. Vaccination coverage and vaccination card usage rates in Nigeria.** Vaccination coverage among children aged 12–23 months in Nigeria and disaggregated by urban and rural areas of residence. Vaccination card coverage is relatively higher in urban areas in comparison to rural areas, and is associated with higher vaccination coverage. Basic vaccination includes 1-dose BCG (Bacille Calmette-Guérin), 3-dose DTP-HepB-Hib (diphtheria, tetanus, pertussis, hepatitis B and *Haemophilus influenzae* type B), 3-dose polio, and 1-dose measles vaccines.

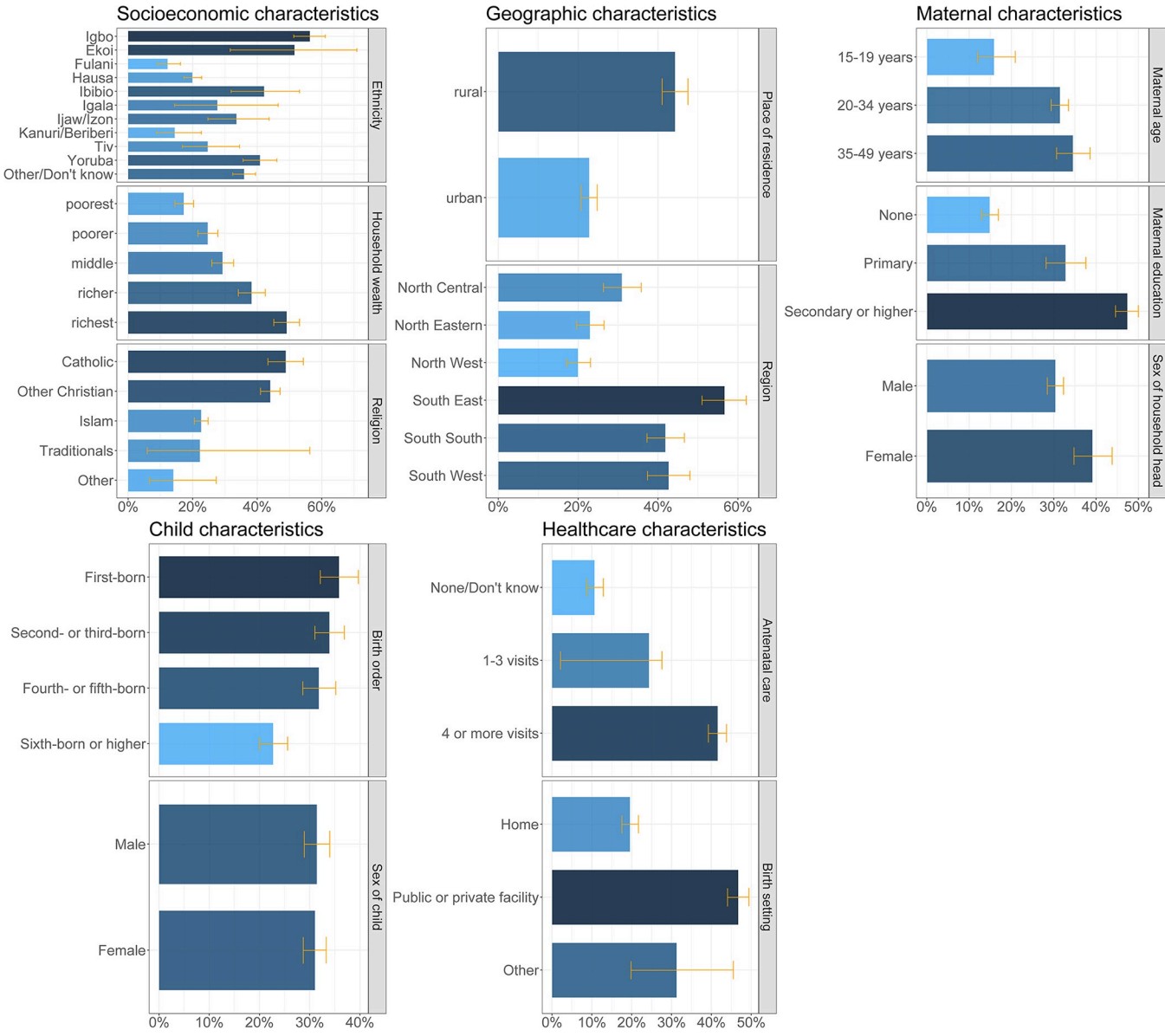

**Fig 3. Basic vaccination coverage in Nigeria.** Basic vaccination coverage among children aged 12–23 months in Nigeria by socioeconomic (household wealth, religion, ethnicity), geographic (region, place of residence), maternal (maternal age at birth, maternal education, maternal household head status), child (sex of child, birth order), and healthcare (birth setting, antenatal care) characteristics.

education, primary education, and secondary education or higher at 15% (13–17%), 33% (28–38%), and 47% (45–50%) respectively. Children living in female-headed households had relatively higher basic vaccination coverage of 39% (35–44%) in comparison to 30% (28–32%) in male-headed households. For child characteristics, basic vaccination coverage was similar among female and male children at 31% (29–33%) and 31% (29–34%) respectively, while coverage decreased by birth order with 36% (32–40%) and 23% (20–26%) among first-born and sixth-born respectively. For healthcare characteristics, children of women who had a higher number of antenatal care visits during their pregnancy had higher basic vaccination coverage at 41% (39–44%) for four or more visits and 11% (8.7–13%) for no or unknown number of visits. More than half of all women gave birth at home, and basic vaccination coverage among

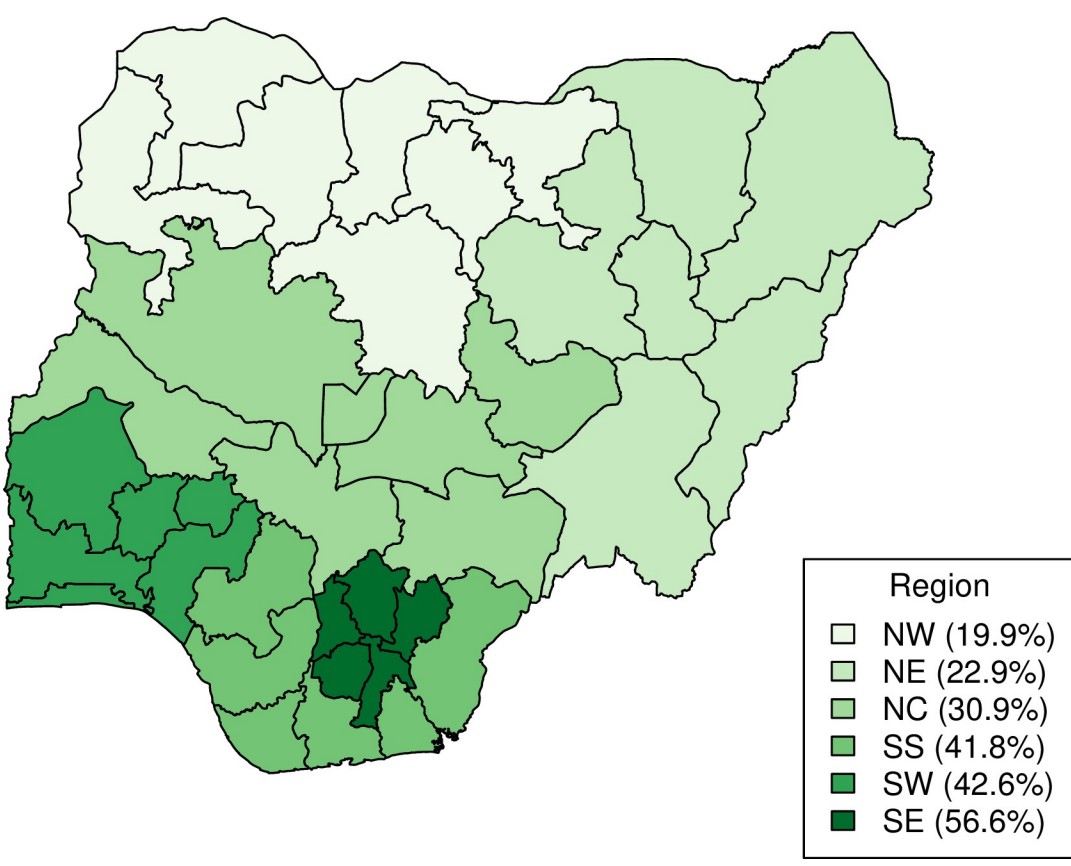

**Fig 4. Basic vaccination coverage in Nigeria at the regional level.** Basic vaccination coverage among children aged 12–23 months in Nigeria at the regional level. (The figure is created by the authors using RStudio and naijR package using data from CIA World Factbook. The figure can be reproduced under CC BY 4.0 license).

these children was relatively lower at 20% (18–22%) in comparison to 47% (44–49%) among children born in a public or private clinical facility.

Fig 5 shows the concentration curve for household wealth-related inequity in basic vaccination coverage, while Table 2 presents the inequities in vaccination card usage rates and vaccination coverage among children aged 12–23 months disaggregated by maternal education, antenatal care, and household wealth in Nigeria, based on Erreygers concentration indices. With respect to maternal education, antenatal care, and household wealth, each of these characteristics had a regressive pro-advantage distribution, with higher vaccination card usage rates and coverage among children from wealthier households, higher maternal education and more antenatal care.

Table 3 and Fig 6 presents the inequities in basic vaccination coverage in Nigeria among children aged 12–23 months associated with socioeconomic (household wealth, religion, ethnicity), geographic (region, place of residence), maternal (maternal age at birth, maternal education, maternal household head status), child (sex of child, birth order), and healthcare (antenatal care, birth setting) characteristics. After controlling for other background characteristics through multiple logistic regression, the adjusted odds ratios (AORs) were significant for the associations between basic vaccination coverage and household wealth, religion, ethnicity, maternal age at birth, maternal education, antenatal care, and birth setting. This model and the AORs include the interaction between household wealth and place of residence (see S3 Table for the strata specific AORs between place of residence and household wealth).

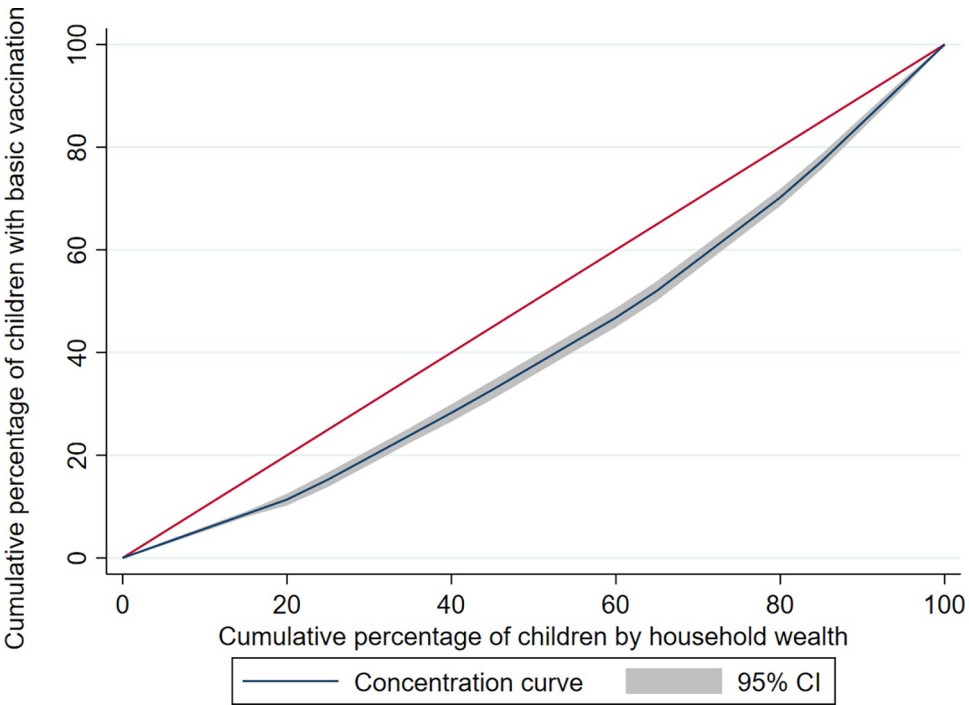

**Fig 5. Wealth-related inequity in basic vaccination coverage in Nigeria.** Concentration curve for household wealth-related inequity in basic vaccination coverage among children aged 12–23 months in Nigeria.

Inequities in basic vaccination coverage (1-dose BCG, 3-dose DTP-HepB-Hib, 3-dose polio, and 1-dose measles) in Nigeria among children aged 12–23 months associated with socioeconomic (household wealth, religion, ethnicity), geographic (region, place of residence), maternal (maternal age at birth, maternal education, maternal household head status), child (sex of child, birth order), and healthcare (antenatal care, birth setting) characteristics. Crude and adjusted odds ratios were estimated using simple and multiple logistic regression respectively.

Children living in households of richest wealth quintiles had 221% higher odds (AOR: 3.21 (2.06, 5.00), p < 0.001) of receiving basic vaccinations in comparison to children of the poorest

**Table 2. Inequities in vaccination card usage rates and vaccination coverage in Nigeria.**

|  | Maternal education | | Antenatal care | | Household wealth | |
|---|---|---|---|---|---|---|
|  | ECI | p-value | ECI | p-value | ECI | p-value |
| Vaccination card | 0.40 | <0.0001 | 0.36 | <0.0001 | 0.30 | <0.0001 |
| Vaccine |  |  |  |  |  |  |
| BCG | 0.48 | <0.0001 | 0.44 | <0.0001 | 0.32 | <0.0001 |
| Measles–first dose | 0.41 | <0.0001 | 0.36 | <0.0001 | 0.30 | <0.0001 |
| DTP-HepB-Hib–third dose | 0.48 | <0.0001 | 0.39 | <0.0001 | 0.31 | <0.0001 |
| Polio–third dose | 0.22 | <0.0001 | 0.22 | <0.0001 | 0.17 | <0.0001 |
| Basic vaccination | 0.32 | <0.0001 | 0.27 | <0.0001 | 0.24 | <0.0001 |

Inequities in vaccination card usage rates and vaccination coverage among children aged 12–23 months disaggregated by maternal education, antenatal care, and household wealth in Nigeria, based on Erreygers concentration indices (ECI). BCG refers to Bacille Calmette-Guérin vaccine and DTP-HepB-Hib refers to diphtheria, tetanus, pertussis (DTP), hepatitis B (HepB) and *Haemophilus Influenzae* type b (Hib).

**Table 3. Inequities in basic vaccination coverage in Nigeria associated with socioeconomic, geographic, maternal, child, and healthcare characteristics.**

| Characteristics | Population (n = 6059) | Mean basic vaccination coverage (% and 95% confidence interval) | Crude odds ratio (OR and 95% confidence interval) | p-value | Adjusted odds ratio (AOR and 95% confidence interval) | p-value |
|---|---|---|---|---|---|---|
| **Socioeconomic** | | | | | | |
| *Household wealth* | | | | | | |
| Poorest | 1301 | 17.2 (14.5, 20.3) | 1 | <0.0001 | 1 | |
| Poorer | 1278 | 24.6 (21.6, 27.8) | **1.57 (1.22, 2.01)** | | 1.38 (0.96, 2.00) | 0.086 |
| Middle | 1239 | 29.2 (25.9, 32.7) | **1.98 (1.53, 2.56)** | | **1.73 (1.17, 2.53)** | **0.005** |
| Richer | 1156 | 38.2 (34.1, 42.5) | **2.97 (2.27, 3.90)** | | **3.10 (2.03, 4.74)** | **<0.001** |
| Richest | 1085 | 49.1 (45.1, 53.1) | **4.64 (3.59, 6.00)** | | **3.21 (2.06, 5.00)** | **<0.001** |
| *Religion* | | | | | | |
| Catholic | 586 | 48.8 (43.3, 54.3) | 1 | <0.0001 | 1 | |
| Other Christian | 1883 | 44.0 (41.0, 47.1) | 0.82 (0.65, 1.04) | | 0.98 (0.76, 1.26) | 0.852 |
| Islam | 3538 | 22.6 (20.5, 24.8) | **0.31 (0.24, 0.39)** | | 0.96 (0.67, 1.38) | 0.823 |
| Traditional | 15 | 22.2 (5.9, 56.3) | 0.30 (0.07, 1.36) | | 0.87 (0.23, 3.25) | 0.830 |
| Other | 37 | 14.0 (6.6, 27.3) | **0.17 (0.07, 0.40)** | | **0.25 (0.10, 0.60)** | **0.002** |
| *Ethnicity* (n = 6057) | | | | | | |
| Igbo | 847 | 56.3 (51.3, 61.1) | 1 | <0.0001 | 1 | |
| Ekoi | 25 | 51.5 (31.6, 70.9) | 0.82 (0.35, 1.94) | | 1.96 (0.74, 5.20) | 0.178 |
| Fulani | 568 | 12.2 (9.0, 16.2) | **0.11 (0.07, 0.16)** | | **0.51 (0.26, 0.97)** | **0.039** |
| Hausa | 1796 | 19.9 (17.3, 22.8) | **0.19 (0.15, 0.25)** | | 0.73 (0.39, 1.36) | 0.324 |
| Ibibio | 105 | 42.1 (31.9, 53.1) | **0.57 (0.35, 0.90)** | | 1.45 (0.77, 2.75) | 0.251 |
| Igala | 55 | 27.6 (14.4, 46.5) | **0.30 (0.13, 0.69)** | | 0.61 (0.24, 1.51) | 0.283 |
| Ijaw/Izon | 171 | 33.5 (24.7, 43.7) | **0.39 (0.24, 0.63)** | | 1.05 (0.54, 2.04) | 0.891 |
| Kanuri/Beriberi | 148 | 14.4 (8.8, 22.7) | **0.13 (0.07, 0.24)** | | 0.42 (0.15, 1.19) | 0.102 |
| Tiv | 148 | 24.6 (16.8, 34.5) | **0.25 (0.15, 0.43)** | | 0.56 (0.27, 1.16) | 0.117 |
| Yoruba | 622 | 40.8 (35.6, 46.1) | **0.53 (0.40, 0.72)** | | 0.70 (0.40, 1.23) | 0.215 |
| Other/Don't know | 1574 | 35.9 (32.4, 39.5) | **0.43 (0.34, 0.56)** | | 1.03 (0.63, 1.67) | 0.912 |
| **Geographic** | | | | | | |
| *Region* | | | | | | |
| North Central | 1061 | 30.9 (26.3, 35.8) | 1 | <0.0001 | 1 | |
| North East | 1303 | 22.9 (19.6, 26.5) | **0.66 (0.50, 0.89)** | | 1.06 (0.75, 1.51) | 0.724 |
| North West | 1697 | 19.9 (17.1, 23.1) | **0.56 (0.42, 0.74)** | | 1.00 (0.67, 1.49) | 1.000 |
| South East | 698 | 56.6 (51.0, 62.1) | **2.93 (2.13, 4.01)** | | 1.35 (0.78, 2.35) | 0.284 |
| South South | 637 | 41.8 (37.2, 46.6) | **1.61 (1.20, 2.16)** | | 0.86 (0.57, 1.27) | 0.441 |
| South West | 663 | 42.6 (37.3, 48.0) | **1.66 (1.22, 2.27)** | | 0.83 (0.56, 1.24) | 0.357 |
| *Place of residence* | | | | | | |
| Urban | 2100 | 44.2 (41.0, 47.5) | 1 | <0.0001 | 1 | |
| Rural | 3959 | 22.7 (20.7, 24.8) | **0.37 (0.31, 0.44)** | | 0.88 (0.56, 1.37) | 0.560 |
| **Maternal** | | | | | | |
| *Maternal age at birth (years)* | | | | | | |
| 15–19 | 366 | 15.8 (11.9, 20.9) | 1 | <0.0001 | 1 | |
| 20–34 | 4330 | 31.4 (29.3, 33.5) | **2.43 (1.72, 3.43)** | | **1.69 (1.15, 2.47)** | **0.007** |
| 35–49 | 1363 | 34.5 (30.6, 38.6) | **2.80 (1.93, 4.05)** | | **2.25 (1.46, 3.49)** | **<0.001** |
| *Maternal education* | | | | | | |
| None | 2614 | 14.8 (12.9, 16.9) | 1 | <0.0001 | 1 | |
| Primary | 881 | 32.7 (28.1, 37.6) | **2.80 (2.15, 3.63)** | | **1.51 (1.14, 1.99)** | **0.004** |
| Secondary or higher | 2564 | 47.3 (44.5, 50.0) | **5.17 (4.27, 6.26)** | | **1.79 (1.39, 2.31)** | **<0.001** |
| *Sex of household head* | | | | | | |

*(Continued)*

**Table 3.** (Continued)

| Characteristics | Population (n = 6059) | Mean basic vaccination coverage (% and 95% confidence interval) | Crude odds ratio (OR and 95% confidence interval) | p-value | Adjusted odds ratio (AOR and 95% confidence interval) | p-value |
|---|---|---|---|---|---|---|
| Male | 5434 | 30.3 (28.4, 32.3) | 1 | <0.0001 | 1 | |
| Female | 625 | 39.1 (34.7, 43.8) | **1.48 (1.21, 1.81)** | | 1.09 (0.86, 1.37) | 0.474 |
| **Child** | | | | | | |
| *Sex* | | | | | | |
| Male | 3148 | 31.4 (28.9, 34.0) | 1 | 0.81 | 1 | |
| Female | 2911 | 31.0 (28.7, 33.3) | 0.98 (0.85, 1.14) | | 1.02 (0.88, 1.19) | 0.766 |
| *Birth order* | | | | | | |
| 1st | 1157 | 35.8 (32.1, 39.7) | 1 | <0.0001 | 1 | |
| 2nd– 3rd | 2081 | 33.9 (31.0, 36.9) | 0.92 (0.75, 1.13) | | 0.91 (0.73, 1.14) | 0.417 |
| 4th– 5th | 1412 | 31.8 (28.6, 35.2) | 0.84 (0.66, 1.06) | | 1.01 (0.79, 1.30) | 0.909 |
| 6th or higher | 1409 | 22.7 (20.0, 25.6) | **0.53 (0.42, 0.65)** | | 0.86 (0.65, 1.14) | 0.292 |
| **Healthcare** | | | | | | |
| *Antenatal care* (n = 5824) | | | | | | |
| None/Don't know | 1529 | 10.6 (8.7, 12.9) | 1 | <0.0001 | 1 | |
| 1–3 visits | 976 | 24.3 (21.3, 27.6) | **2.70 (2.07, 3.53)** | | **2.15 (1.62, 2.85)** | **<0.001** |
| 4 or more visits | 3319 | 41.5 (39.2, 43.8) | **5.96 (4.71, 7.53)** | | **2.87 (2.21, 3.72)** | **<0.001** |
| *Birth setting* | | | | | | |
| Home setting | 3415 | 19.5 (17.5, 21.7) | 1 | <0.0001 | 1 | |
| Public or private facility | 2549 | 46.7 (44.0, 49.4) | **3.61 (3.06, 4.26)** | | **1.32 (1.08, 1.61)** | **0.006** |
| Other | 95 | 31.2 (19.8, 45.5) | 1.87 (0.98, 3.56) | | 0.77 (0.40, 1.50) | 0.444 |

Bold values represent the statistically significant results.

wealth quintiles. However, children living in rural areas in the richer quintile had 64% lower odds (AOR: 0.36 (0.24, 0.53), p < 0.001) and children in rural areas in the richest quintile 50% lower odds (AOR: 0.50 (0.35, 0.72), p <0.001) than children in urban areas. Children whose religion was classified as 'other' had 75% lower odds (AOR: 0.25 (0.10, 0.60), p < 0.002) of receiving basic vaccinations than children of Catholic faith. In comparison to children of Igbo ethnicity, children of Fulani ethnicity had 49% lower odds of receiving basic vaccinations (AOR: 0.51 (0.26, 0.97), p = 0.039). Children of mothers aged 35–49 years had 125% higher odds (AOR: 2.25 (1.46, 3.49), p < 0.001) of receiving basic vaccination than children of mothers aged 15–19 years. Children of mothers with secondary or higher education had 79% higher odds (AOR: 1.79 (1.39, 2.31), p < 0.001) of receiving basic vaccination in comparison to children of mothers with no formal education. Children of mothers who had four or more antenatal care visits had 187% higher odds (AOR: 2.87 (2.21, 3.72), p < 0.001) of receiving basic vaccinations than children of mothers who had no antenatal care. Children born in clinical facilities had 32% higher odds (AOR 1.32 (1.08, 1.61), p = 0.006) than children born in home settings of receiving basic vaccinations.

## Discussion

We conducted a systematic review to identify the social determinants of childhood immunisation in low- and middle-income countries. We selected household wealth, religion, and ethnicity for socioeconomic characteristics; region and place of residence for geographic characteristics; maternal age at birth, maternal education, and maternal household head status for maternal characteristics; sex of child and birth order for child characteristics; and antenatal

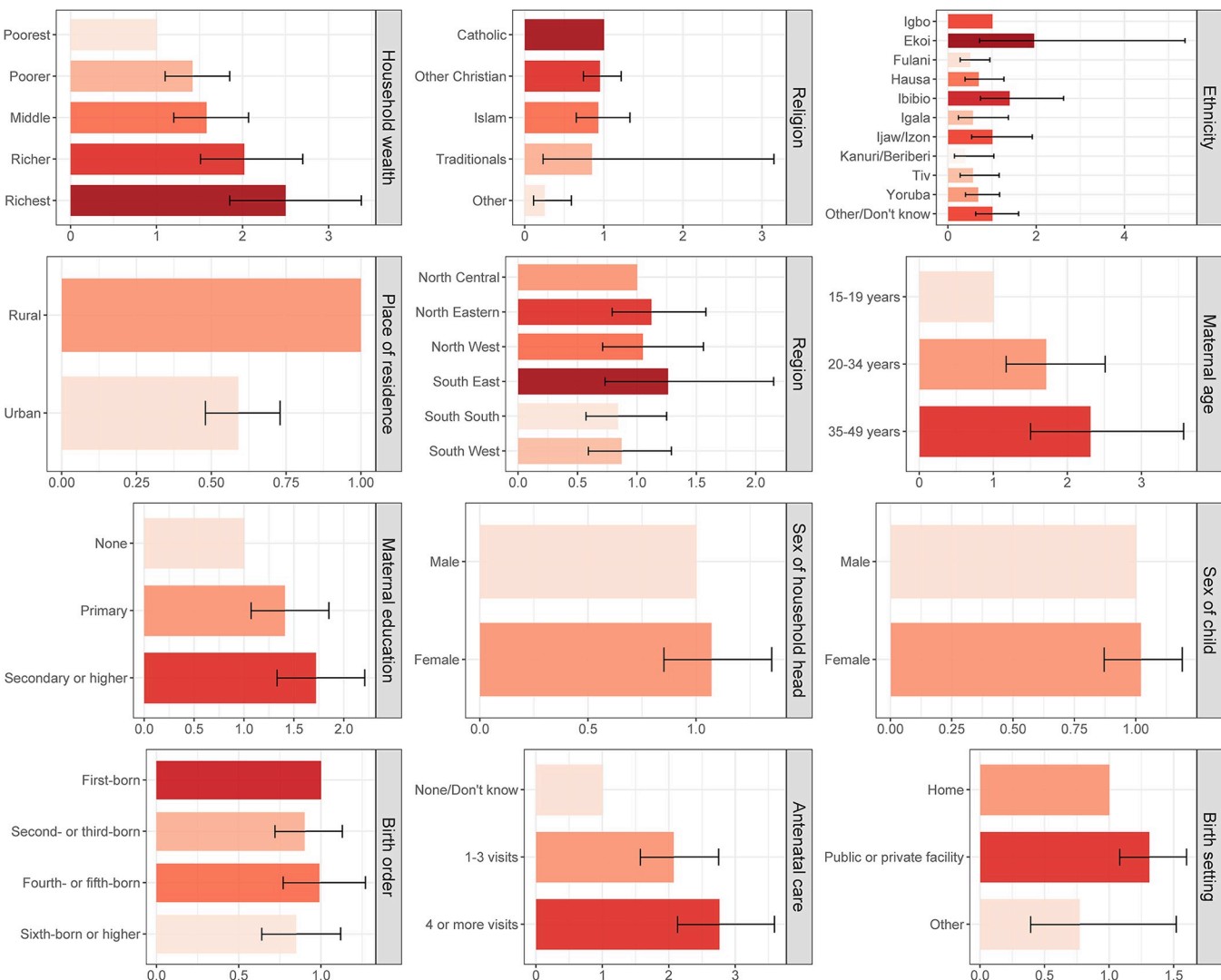

**Fig 6. Inequities in basic vaccination coverage in Nigeria.** Inequities in basic vaccination coverage among children aged 12–23 months in Nigeria associated with socioeconomic (household wealth, religion, ethnicity), geographic (region, place of residence), maternal (maternal age at birth, maternal education, maternal household head status), child (sex of child, birth order), and healthcare (birth setting, antenatal care) characteristics.

care and birth setting for healthcare characteristics. Based on the characteristics identified from the systematic review, we applied a social determinants framework to assess basic vaccination coverage (1-dose BCG, 3-dose DTP-HepB-Hib, 3-dose polio, and 1-dose measles) among children aged 12–23 months in Nigeria using the 2018 Nigeria DHS survey dataset.

The associations identified in this study between basic vaccination coverage and socioeconomic, geographic, maternal, child, and healthcare characteristics identified are supported by other studies. Basic vaccination coverage was associated with household wealth, and families in the richest quintile are more likely to live in urban areas with better access to functional private and public health facilities that provide immunisation services [24, 80]. Mothers in urban areas are more likely to use preventive healthcare services, including childhood immunisation, due to their proximity to healthcare facilities in urban settings and higher educational status in comparison to mothers in rural areas with higher travel costs [81–83]. Children in the richer households of rural areas had reduced odds of basic vaccination coverage in comparison to

children in the poorest households of urban areas. Children of Fulani ethnicity had lower odds of basic vaccination compared to Igbo children, and there is evidence that awareness of immunisation is low amongst Fulani mothers [84, 85]. Further, Fulani ethnic groups reside in mostly rural settings and are nomadic, limiting their access to health services including immunisation services.

We found that basic vaccination coverage was associated with maternal education. Educated mothers have better awareness and knowledge on childhood immunisation and are more able to overcome cultural barriers to vaccination [86]. Maternal age was associated with basic vaccination coverage and older mothers may have more experience with antenatal clinics and have greater awareness of immunisation services from previous children [26]. They are also more likely to have financial access to immunisation services and live in urban settings with improved access to immunisation services.

Delivery in a health facility and antenatal care were associated with basic vaccination, and to give birth in a health facility indicates that mothers have overcome barriers to accessing health services, and mothers will also receive information on childhood immunisation from healthcare workers there. Hence, utilisation of health services by mothers leads to improved immunisation status of their children [87].

At the regional level, basic vaccination among children ranged from the lowest coverage in the North West region to nearly three times higher coverage in the South East region, although residents in Northern Nigeria are more likely to have immunisation services within 5 km [59]. The large degree of autonomy of different states and the impact of ongoing conflict in parts of the country, in addition to socio-cultural reasons, can explain in part the geographical disparity in immunisation services [14]. However, regional differences in coverage were not significant in the multivariable logistic regression model.

Despite vaccination being provided at no cost to individuals, coverage was still below target levels for even the most advantaged groups and therefore we recommend a proportionate universalism approach with actions proportionate to the level of disadvantage [88, 89]. There is some evidence of payment being required for vaccination, even in public health facilities, and this may threaten vaccination uptake [82]. Both the oversight of immunisation services and public awareness of vaccinations' no cost status should be strengthened.

Higher coverage with more antenatal care indicates greater engagement with health services and thereby providing more opportunities for vaccine education. The integration of immunisation services to nutrition programmes and paediatric outpatient departments of primary healthcare centres has been shown to improve coverage and decrease drop-out rates in South Sudan [90]. Hence, we recommend efforts to engage with families during health care visits outside of routine immunisation services and to take advantage of these missed opportunities to reach under-immunised children [91].

Our estimates of vaccination coverage in Nigeria for 2018 were lower than national estimates based on administrative reporting from health service providers, though closer to WHO and UNICEF Estimates of National Immunization Coverage (WEUNIC) (see S4 Table) [92]. The coverage gaps have been shown to be systematically underestimated by administrative reporting in Nigeria [93]. In 80% of Nigerian states, the basic vaccination coverage was below the national target of 95% or higher coverage and below the Gavi target of 90% DTP3 coverage [94]. The below target coverage for first doses indicates challenges with access to immunisation services while the decrease in coverage for subsequent doses indicates drop-out due to insufficient knowledge on dose completion [9, 95, 96]. The COVID-19 pandemic has disrupted vaccination globally and coverage was lower than expected in Nigeria in 2020 [17, 18, 97], highlighting the urgent need for catch-up vaccination to close the immunity gaps and prevent vaccine-preventable disease outbreaks [98]. The role of conflict on these disparities is also

important to understand. A literature review of conflict and vaccination inferred that conflict-affected countries had vaccination coverage below global levels and conflict affected vaccination services and human resources, including attacks on healthcare workers, which has happened in Nigeria [14, 70]. Our study inferences complement the findings in related studies in Nigeria and other countries.

The proportion of zero-dose children was high and raises the risk of vaccine-preventable disease outbreaks, with nearly 25% of children in rural areas not receiving a single dose of any of the basic vaccinations. Zero-dose children are also likely to lack access to health and welfare services and to suffer from multiple sources of deprivation [99, 100]. More than 50% of children do not have vaccination cards and nearly 60% of children do not have birth registrations [101] - improving uptake of vaccination cards and birth registrations would help in part to address the barriers for vaccine access for all children, including zero-dose children.

Our study has limitations, and we cannot estimate causal-effect relationships nor temporal inferences due to the cross-sectional study design of 2018 Nigeria DHS. Our study has similar biases that are associated with DHS surveys, including recall bias, measurement bias, and social desirability bias which tend to overestimate vaccination coverage. In particular, only 49% of children had a vaccination card and when a vaccination card was not available for a child, their mother was asked to recall their vaccinations, which may lead to an overestimation of coverage.

For future work, we recommend qualitative research to understand the barriers and enablers of childhood vaccination and their associations with socioeconomic, geographic, maternal, child, and healthcare characteristics which would be valuable to adapt vaccination programmes to improve coverage equitably in Nigeria.

## Conclusions

We identified the inequities in basic vaccination coverage by socioeconomic, geographic, maternal, child, and healthcare characteristics among children aged 12–23 months in Nigeria using a social determinants of health perspective. In conclusion, we infer that inequities in basic vaccination were associated with lower coverage among children living in poorer households, belonging to Fulani ethnicity, born in home settings in Nigeria, with younger mothers at birth, with mothers with no formal education and with mothers who had no antenatal care visits. We recommend a proportionate universalism approach with targeted vaccination programmes proportionate to the level of disadvantage for addressing the immunisation barriers faced by these underserved subpopulations. This will improve coverage and reduce inequities in childhood immunisation associated with socioeconomic, geographic, maternal, child, and healthcare characteristics in Nigeria.

## Supporting information

**S1 Checklist. PRISMA checklist.**
(DOCX)

**S1 Table. Routine vaccination schedule of under 1-year-old children in Nigeria.**
(DOCX)

**S2 Table. Vaccination coverage and vaccination card usage rates in Nigeria.**
(DOCX)

**S3 Table. Interaction effects.**
(DOCX)

**S4 Table. Vaccine coverage.** Vaccine coverage estimates for Nigeria in 2018 based on DHS, WUENIC (WHO-UNICEF), administrative, and official country sources.
(DOCX)

**S1 Fig. PRISMA flowchart.** The Preferred Reporting Items for Systematic Reviews and Meta Analyses (PRISMA) flow diagram of articles' identification, screening, eligibility, and inclusion in the systematic review is illustrated.
(DOCX)

## Acknowledgments

We thank the DHS Program for access to the 2018 Nigeria Demographic and Health Survey dataset.

## Author Contributions

**Conceptualization:** Sarah V. Williams, Kaja Abbas.

**Formal analysis:** Sarah V. Williams, Tanimola Akande, Kaja Abbas.

**Supervision:** Kaja Abbas.

**Writing – original draft:** Sarah V. Williams, Kaja Abbas.

**Writing – review & editing:** Sarah V. Williams, Tanimola Akande, Kaja Abbas.

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
