## [Decision Letter · Decision Letter 0]

13 Mar 2023

PONE-D-23-02804Childhood vaccination coverage by socioeconomic, geographic, maternal, child, and healthcare characteristics in Nigeria: equity impact analysisPLOS ONE

Dear Dr. Williams,

Thank you for submitting your manuscript to PLOS ONE. After careful consideration, we feel that it has merit but does not fully meet PLOS ONE’s publication criteria as it currently stands. Therefore, we invite you to submit a revised version of the manuscript that addresses the points raised during the review process.

We look forward to receiving your revised manuscript.

Kind regards,

Harapan Harapan, MD, PhD

Academic Editor

PLOS ONE

Journal Requirements:

"KA is supported by the Vaccine Impact Modelling Consortium (OPP1157270). 

" ext-link-type="uri" xlink:type="simple">https://www.vaccineimpact.org/"

4. We note that Figure 4 in your submission contain map image which may be copyrighted. All PLOS content is published under the Creative Commons Attribution License (CC BY 4.0), which means that the manuscript, images, and Supporting Information files will be freely available online, and any third party is permitted to access, download, copy, distribute, and use these materials in any way, even commercially, with proper attribution. For these reasons, we cannot publish previously copyrighted maps or satellite images created using proprietary data, such as Google software (Google Maps, Street View, and Earth). For more information, see our copyright guidelines: http://journals.plos.org/plosone/s/licenses-and-copyright.

a. You may seek permission from the original copyright holder of Figure 4 to publish the content specifically under the CC BY 4.0 license.  

Reviewers' comments:

Reviewer's Responses to Questions

**Comments to the Author**

1. Is the manuscript technically sound, and do the data support the conclusions?

Reviewer #1: Partly

Reviewer #2: Yes

2. Has the statistical analysis been performed appropriately and rigorously? 

Reviewer #1: No

Reviewer #2: Yes

3. Have the authors made all data underlying the findings in their manuscript fully available?

Reviewer #1: Yes

Reviewer #2: Yes

4. Is the manuscript presented in an intelligible fashion and written in standard English?

Reviewer #1: Yes

Reviewer #2: Yes

5. Review Comments to the Author

Reviewer #1: Thank you for the opportunity of reviewing “Childhood vaccination coverage by socioeconomic, geographic, maternal, child, and healthcare characteristics in Nigeria: equity impact analysis” . The study involved 6,059 children (12-23 months old), where the data are based from the 2018 Nigeria Demographic and Health Survey. The study revealed that, in Nigeria, the basic vaccination coverage reached 31% (95% CI: 29–33), while 19% (95% CI:18–21) of children aged 12-23 months had received no basic vaccines. Determinants such as household wealth, religion, and ethnicity, region and place of residence, maternal age at birth, maternal education, and maternal household head status, sex of child, and birth order, antenatal care, and birth setting. Several comments:

1. I find this manuscript being hard to read. The structure is confusing and hard to follow.

2. Abstract is lack of quantitative results. Please state the number instead of what have been calculated.

3. I don’t understand the term ‘equity’ the author put in the title. Why the study is different to other study who report the determinants of vaccine coverage?

4. Please add “Study design” section in the “Methods”. Also, please disclose the time of the study therein.

5. Please improve the table presentation.

6. The figures are not well organized.

7. In introduction, please highlight the urgency of this study reporting the data from a low-income country. A previous study has hinted that the low- and middle-income countries suffer the most from the lack of vaccine coverage (https://narraj.org/main/article/view/57 and others).

8. I understand that the study was based on 2018 survey. But, could you give a comment on how the current pandemic could further hamper the vaccination inequity. (refer to: https://narraj.org/main/article/view/7 or https://doi.org/10.7189%2Fjogh.11.03086)

Reviewer #2: As the issue of vaccine inequity in low- and middle-income countries is still one of public health concerns, I think this paper is very interesting and important.

The authors have done a wonderful job by clearly stating the aim of the study, the significance as well as the importance of this research. The methods section was presented clearly, so that it was easy to follow each step of the investigation. The results were presented in an intangible manner, and the discussion covers all important findings in the study. Well done!

However, there is one thing I would like to clarify. In the results section, line 220-222, it was written that “Vaccination cards were available for 49% (46-51%) of children, among which 57% (55-60%) had received all basic vaccinations. Among the 51% (49-54%) of children without vaccination cards, only 6.4% (5.3-7.7%) of them had received all basic vaccinations.”

My question is, at what age does the complete vaccination program should be accomplished? As the population of this study was children aged 12-23 months, there might be possibility that children who have not received all basic vaccinations were younger than the cut off for complete vaccination (explaining why they have not completed all basic vaccinations). How will the authors explain this result?

Above all, I think the authors have done an incredible job in presenting such important study, which helpfully will positively contribute to better vaccination coverage in Nigeria.

6. PLOS authors have the option to publish the peer review history of their article (what does this mean?). If published, this will include your full peer review and any attached files.

Reviewer #1: No

Reviewer #2: **Yes: **Amanda Yufika, MD, MSc

---

## [Author Response · Author response to Decision Letter 0]

14 May 2023

Response to Academic Editor comments

1. Comment: Please ensure that your manuscript meets PLOS ONE's style requirements, including those for file naming. The PLOS ONE style templates can be found at 

Response: The manuscript has been updated to follow the PLOS ONE’s style requirements.

2. Comment: Please provide an amended statement that declares *all* the funding or sources of support (whether external or internal to your organization) received during this study, as detailed online in our guide for authors at http://journals.plos.org/plosone/s/submit-now. 

Response: We have updated the funding statement

KA is supported by the Vaccine Impact Modelling Consortium (OPP1157270). There was no additional external funding received for this study. The authors declare that they have no known competing financial interests or personal relationships that could have appeared to influence the work reported in this paper.

3. Comment: Your ethics statement should only appear in the Methods section of your manuscript. 

Response: We have moved the ethics statement to the Methods section

4. Comment: We note that Figure 4 in your submission contains a map image which may be copyrighted. We require you to either (1) present written permission from the copyright holder to publish these figures specifically under the CC BY 4.0 license, or (2) remove the figures from your submission

Response: This figure has been redrawn using RStudio and naijR package using data from the CIA World Factbook. The figure caption has been updated as below:

(The figure is created by the authors using RStudio and naijR package using data from CIA World Factbook. The figure can be reproduced under CC BY 4.0 license) 

5. Comment: Please include captions for your Supporting Information files at the end of your manuscript, and update any in-text citations to match accordingly. Please see our Supporting Information guidelines for more information: http://journals.plos.org/plosone/s/supporting-information. 

Response: We have updated the captions and in-text citations

Response to Reviewer Comments

Reviewer #1

● Comment: Thank you for the opportunity of reviewing “Childhood vaccination coverage by socioeconomic, geographic, maternal, child, and healthcare characteristics in Nigeria: equity impact analysis” . The study involved 6,059 children (12-23 months old), where the data are based from the 2018 Nigeria Demographic and Health Survey. The study revealed that, in Nigeria, the basic vaccination coverage reached 31% (95% CI: 29–33), while 19% (95% CI:18–21) of children aged 12-23 months had received no basic vaccines. Determinants such as household wealth, religion, and ethnicity, region and place of residence, maternal age at birth, maternal education, and maternal household head status, sex of child, and birth order, antenatal care, and birth setting. Several comments:

Response: We thank you for your valuable feedback. We have updated the manuscript to address your comments, and the updates in the paper are highlighted with yellow background colour. 

1. Comment: I find this manuscript being hard to read. The structure is confusing and hard to follow.

Response: We have updated the paper to improved the readability.

2. Comment: Abstract is lack of quantitative results. Please state the number instead of what have been calculated.

Response: We have added quantitative results of adjusted odds ratios to the Abstract as follows: 

Results: From the systematic review, we identified the key determinants of immunisation to be household wealth, religion, and ethnicity for socioeconomic characteristics; region and place of residence for geographic characteristics, maternal age at birth, maternal education, and household head status for maternal characteristics; sex of child and birth order for child characteristics; and antenatal care and birth setting for healthcare characteristics. Based on the 2018 Nigeria DHS analysis of 6,059 children aged 12-23 months, we estimated that basic vaccination coverage was 31% (95% CI: 29–33) among children aged 12-23 months, whilst 19% (95% CI:18–21) of them were zero-dose children who had received none of the basic vaccines. After controlling for background characteristics, there was a significant increase in the odds of basic vaccination by household wealth (AOR: 3.21 (2.06, 5.00), p 0.001) for the wealthiest quintile compared to the poorest quintile, antenatal care of four or more antenatal care visits compared to no antenatal care (AOR: 2.87 (2.21, 3.72), p 0.001), delivery in a health facility compared to home births (AOR 1.32 (1.08, 1.61), p = 0.006), relatively older maternal age of 35-49 years compared to 15-19 years (AOR: 2.25 (1.46, 3.49), p 0.001), and maternal education of secondary or higher education compared to no formal education (AOR: 1.79 (1.39, 2.31), p 0.001). Children of Fulani ethnicity in comparison to children of Igbo ethnicity had lower odds of receiving basic vaccinations (AOR: 0.51 (0.26, 0.97), p = 0.039). 

3. Comment: I don’t understand the term ‘equity’ the author put in the title. Why the study is different to other study who report the determinants of vaccine coverage?

Response: In the fourth paragraph of Introduction section, we have explained why we refer to inequity rather than inequality. 

We refer to vaccine inequity as unfair and avoidable or remediable differences in health among population groups defined socially, economically, demographically, or geographically [21]. This is related to but distinct from health inequality, which indicates the status of imbalances or differences in health among population groups without any moral judgement on whether the imbalances or differences are fair or not [22,23].

4. Comment: Please add “Study design” section in the “Methods”. Also, please disclose the time of the study therein.

Response: We have updated the title of the below section in the Methods which states the study time – “We analysed the 2018 Nigeria DHS which was conducted between August to December 2018 [74]. “

Study design of demographic and health survey

We analysed the 2018 Nigeria DHS which was conducted between August to December 2018 [74]. The DHS are nationally representative household surveys focusing on population, health, and nutrition in LMICs [75]. The DHS sample is a two-stage stratified cluster sample with sampling weights applied to ensure that results are representative. There are four questionnaires: household questionnaire, woman’s questionnaire, man’s questionnaire, and biomarker questionnaire. The country is divided into clusters with 30 households selected from each cluster. The woman’s questionnaire was asked to women aged 15-49 years and provides the data for our study. All women aged 15-49 years in the sampled households were included and the survey was successfully conducted in 1,389 clusters after 11 clusters were dropped following deteriorating security in those areas during data collection. In addition, in the state of Borno, 11 of the 27 Local Government Areas (LGAs) in the state were dropped due to insecurity. Clusters selected from these dropped LGAs were replaced with clusters from the remaining LGAs and so may not be representative of the entire state [74]. The study population for our analysis was women aged 15-49 years with a child aged 12 to 23 months old. Of the 41,821 women interviewed, 33,924 women had a child aged 59 months or younger and immunisation data was collected for 6,059 living children aged between 12-23 months. We applied sampling weights to the survey dataset to adjust for disproportionate sampling and non-response, thereby ensuring that the sample was representative of the population.

5. Comment: Please improve the table presentation.

Response: Thank you – we have presented the table with all the required information and this will be formatted as per the journal’s requirements during typesetting. 

6. Comment: The figures are not well organized.

Response: We have uploaded high-resolution figures as separate file. The figures will be organised as per the journal’s requirements during typesetting. 

7. Comment: In introduction, please highlight the urgency of this study reporting the data from a low-income country. A previous study has hinted that the low- and middle-income countries suffer the most from the lack of vaccine coverage (https://narraj.org/main/article/view/57 and others).

Response: The third paragraph in the Introduction includes information that Nigeria has the most under-immunised children in the world with 4.5 million in 2018, highlighting the immunisation system challenges. We have updated this to include the immunisation challenges from COVID-19 and have referenced the study from comment 8 in the third paragraph in the Introduction section: https://narraj.org/main/article/view/7. 

This study was included out of context – Hassan W, Kazmi SK, Tahir MJ, Ullah I, Royan HA, Fahriani M, et al. Global acceptance and hesitancy of COVID-19 vaccination: A narrative review. Narra J. 2021;1. doi:10.52225/narra.v1i3.57. 

The Expanded Programme on Immunisation (EPI) was established by the World Health Organization (WHO) in 1974 to improve vaccination services globally [12], and Nigeria began nationwide implementation of EPI in 1979 which was later changed to the National Programme on Immunization [13]. Although the vaccines in the routine immunisation programme (Table S1) for under 5-year-old children are available with no out-of-pocket charges [14], Nigeria has the most under-immunised children in the world with 4.5 million in 2018 [15]. The immunisation system challenges in Nigeria include weak institutions, service delivery, funding, infrastructure, poor coordination between the National Programme on Immunization and non-governmental organisations delivering vaccination services, and a lack of political commitment in some regions, with further challenges to immunisation services caused by the COVID-19 pandemic [14,16–18]. There are fewer adequately skilled healthcare personnel in rural areas and northern states, and poor retention and frequent transfers of workers. Security is also an issue, with attacks on healthcare workers in recent years. Attitudes of communities and caregivers are important too, with a lack of knowledge about vaccination and mistrust of services hindering vaccination uptake [16,19].

8. Comment: I understand that the study was based on 2018 survey. But, could you give a comment on how the current pandemic could further hamper the vaccination inequity. (refer to: https://narraj.org/main/article/view/7 or https://doi.org/10.7189%2Fjogh.11.03086)

Response: The ninth paragraph in the Discussion section highlights the COVID-19 disruption on routine immunisation. The two references are now included in this paragraph. 

Our estimates of vaccination coverage in Nigeria for 2018 were lower than national estimates based on administrative reporting from health service providers, though closer to WHO and UNICEF Estimates of National Immunization Coverage (WEUNIC) (see Appendix S5) [92]. The coverage gaps have been shown to be systematically underestimated by administrative reporting in Nigeria [93]. In 80% of Nigerian states, the basic vaccination coverage was below the national target of 95% or higher coverage and below the Gavi target of 90% DTP3 coverage [94]. The below target coverage for first doses indicates challenges with access to immunisation services while the decrease in coverage for subsequent doses indicates drop-out due to insufficient knowledge on dose completion [9,95,96]. The COVID-19 pandemic has disrupted vaccination globally and coverage was lower than expected in Nigeria in 2020 [17,18,97], highlighting the urgent need for catch-up vaccination to close the immunity gaps and prevent vaccine-preventable disease outbreaks [98]. The role of conflict on these disparities is also important to understand. A literature review of conflict and vaccination inferred that conflict-affected countries had vaccination coverage below global levels and conflict affected vaccination services and human resources, including attacks on healthcare workers, which has happened in Nigeria [14,64]. Our study inferences complement the findings in related studies in Nigeria and other countries (see Table 1).

 

Reviewer #2

● Comment: As the issue of vaccine inequity in low- and middle-income countries is still one of public health concerns, I think this paper is very interesting and important.

 The authors have done a wonderful job by clearly stating the aim of the study, the significance as well as the importance of this research. The methods section was presented clearly, so that it was easy to follow each step of the investigation. The results were presented in an intangible manner, and the discussion covers all important findings in the study. Well done!

○ Response: We thank you for your valuable feedback. We have updated the manuscript to address your comments, and the updates in the paper are highlighted with yellow background colour. 

● Comment: However, there is one thing I would like to clarify. In the results section, line 220-222, it was written that “Vaccination cards were available for 49% (46-51%) of children, among which 57% (55-60%) had received all basic vaccinations. Among the 51% (49-54%) of children without vaccination cards, only 6.4% (5.3-7.7%) of them had received all basic vaccinations.”

 My question is, at what age does the complete vaccination program should be accomplished? As the population of this study was children aged 12-23 months, there might be possibility that children who have not received all basic vaccinations were younger than the cut off for complete vaccination (explaining why they have not completed all basic vaccinations). How will the authors explain this result?

○ Response: The routine vaccinations in Nigeria are listed in S1 Table in the appendix. By 9 months children should have received all of their basic vaccinations and so children included in the study (aged 12-23 months) should have received all of their basic vaccinations by this age.

● Comment: Above all, I think the authors have done an incredible job in presenting such important study, which helpfully will positively contribute to better vaccination coverage in Nigeria.

○ Response: Thank you for your valuable feedback.

---

## [Decision Letter · Decision Letter 1]

6 Jun 2023

PONE-D-23-02804R1Childhood vaccination coverage by socioeconomic, geographic, maternal, child, and healthcare characteristics in Nigeria: equity impact analysisPLOS ONE

Dear Dr. Williams,

Thank you for submitting your manuscript to PLOS ONE. After careful consideration, we feel that it has merit but does not fully meet PLOS ONE’s publication criteria as it currently stands. Therefore, we invite you to submit a revised version of the manuscript that addresses the points raised during the review process.

We look forward to receiving your revised manuscript.

Kind regards,

Harapan Harapan, MD, PhD

Academic Editor

PLOS ONE

Journal Requirements:

Reviewers' comments:

Reviewer's Responses to Questions

**Comments to the Author**

1. If the authors have adequately addressed your comments raised in a previous round of review and you feel that this manuscript is now acceptable for publication, you may indicate that here to bypass the “Comments to the Author” section, enter your conflict of interest statement in the “Confidential to Editor” section, and submit your "Accept" recommendation.

Reviewer #1: (No Response)

Reviewer #2: All comments have been addressed

2. Is the manuscript technically sound, and do the data support the conclusions?

Reviewer #1: Yes

Reviewer #2: Yes

3. Has the statistical analysis been performed appropriately and rigorously? 

Reviewer #1: Yes

Reviewer #2: Yes

4. Have the authors made all data underlying the findings in their manuscript fully available?

Reviewer #1: Yes

Reviewer #2: Yes

5. Is the manuscript presented in an intelligible fashion and written in standard English?

Reviewer #1: Yes

Reviewer #2: Yes

6. Review Comments to the Author

Reviewer #1: There are some minor and technical issues left. Please find them below:

1. Thank you for the clarification regarding “inequality”. Nonetheless, I still could not see what is new from this study. In introduction. please tell explicitly what have been performed in the past studies, and what is new in this present study.

2. Study design should be put before the “Characteristics selection and systematic review”. The section details why systematic review performed and how the information is used in the following study. This section will ease reader to understand the general overview of the methods before going deeper to details.`

3. Consider to move Table 1 to results.

4. In tables. “Probability” do you mean p-value? Also please define all abbreviation in table footnote including “DTP-HepB-Hib” and many others.

5. Table 3. Some values were written in bold – not sure why. Please explain why they were presented in bold in table footnote.

Reviewer #2: I would like to thank the authors for addressing my comments. No more comments or suggestions from me. All the best with the manuscript.

7. PLOS authors have the option to publish the peer review history of their article (what does this mean?). If published, this will include your full peer review and any attached files.

Reviewer #1: No

Reviewer #2: No

---

## [Author Response · Author response to Decision Letter 1]

19 Jun 2023

Second revision (June 6, 2023)

Response to Reviewer Comments

Reviewer #1

● Comment: There are some minor and technical issues left. Please find them below:

Response: We thank you for your valuable feedback. We have updated the manuscript to address your comments, and the updates in the paper are highlighted with yellow background colour. 

1. Comment: Thank you for the clarification regarding “inequality”. Nonetheless, I still could not see what is new from this study. In introduction. please tell explicitly what have been performed in the past studies, and what is new in this present study.

Response: 

● At the end of Introduction section, we have added the sentence – “We conducted disaggregated equity impact analysis to reveal the inequities in basic vaccination coverage that are hidden at the aggregated national level, and understand the facilitators and barriers to vaccination through the social determinants of health framework. ”

● Throughout the Discussion section, we highlight the reasons for the inferences from our analysis with supporting evidence from related studies in Nigeria and other countries as applicable. However, this differentiates our study because in addition to identifying characteristics associated with basic vaccination coverage, we highlight the facilitators and barriers faced by different social subgroups specifically in the Nigerian context. 

2. Comment: Study design should be put before the “Characteristics selection and systematic review”. The section details why systematic review performed and how the information is used in the following study. This section will ease reader to understand the general overview of the methods before going deeper to details.`

Response: We have moved the subsection “Study design of demographic and health survey” before the subsection “Characteristics selection and systematic review”. 

3. Comment: Consider to move Table 1 to results.

Response: We have moved Table 1 to results. 

4. Comment: In tables. “Probability” do you mean p-value? Also please define all abbreviation in table footnote including “DTP-HepB-Hib” and many others.

Response: 

● Yes, “Probability” mean p-value. We have replaced “probablity” with “p-value” in Table 2 column header. 

● We have added the footnote to tables – BCG refers to Bacille Calmette-Guérin and DTP-HepB-Hib refers to diphtheria, tetanus, pertussis (DTP), hepatitis B (HepB) and Haemophilus Influenzae type b (Hib).

5. Comment: Some values were written in bold – not sure why. Please explain why they were presented in bold in table footnote.

Response: 

● We have added a note at the bottom of Table 3 – Bold values represent the statistically significant results.

---

## [Editor Report · Decision Letter 2]

27 Oct 2023

PONE-D-23-02804R2Childhood vaccination coverage by socioeconomic, geographic, maternal, child, and healthcare characteristics in Nigeria: equity impact analysisPLOS ONE

Dear Dr. Williams,

Thank you for submitting your manuscript to PLOS ONE. After careful consideration, we feel that it has merit but does not fully meet PLOS ONE’s publication criteria as it currently stands. Therefore, we invite you to submit a revised version of the manuscript that addresses the points raised during the review process.

We look forward to receiving your revised manuscript.

Kind regards,

Steve Zimmerman, PhD

Senior Editor, PLOS ONE

on behalf of

Harapan Harapan, MD, PhD

Academic Editor

PLOS ONE

Journal Requirements:

**Additional Editor Comments:**

Thank you for submitting your manuscript to PLOS ONE. After careful consideration, we feel that it has satisfied our scientific requirements for publication.

However, we also note that this manuscript contains a systematic review; our author guidelines require that you upload a PRISMA checklist as supporting information (https://journals.plos.org/globalpublichealth/s/submission-guidelines#loc-systematic-reviews-and-meta-analyses). Information about the PRISMA guidance and blank checklists can be found here: http://www.prisma-statement.org/.

Please could you resubmit with a completed copy of the PRISMA checklist as a "Supporting Information" file?

---

## [Author Response · Author response to Decision Letter 2]

13 Dec 2023

We have submitted a PRISMA checklist as a supporting information file and alongside this we would like to revise the title of the manuscript to, “Systematic review of social determinants of childhood immunisation in low- and middle-income countries and equity impact analysis of childhood vaccination coverage in Nigeria” so that readers are aware that it is a systematic review. We have updated the manuscript with changes tracked.

---

## [Editor Report · Decision Letter 3]

4 Jan 2024

Systematic review of social determinants of childhood immunisation in low- and middle- income countries and equity impact analysis of childhood vaccination coverage in Nigeria

PONE-D-23-02804R3

Dear Dr. Williams,

We’re pleased to inform you that your manuscript has been judged scientifically suitable for publication and will be formally accepted for publication once it meets all outstanding technical requirements.

Kind regards,

Harapan Harapan, MD, PhD

Academic Editor

PLOS ONE
---

## [Editor Report · Acceptance letter]

27 Feb 2024

PONE-D-23-02804R3 

PLOS ONE

Dear Dr. Williams, 

I'm pleased to inform you that your manuscript has been deemed suitable for publication in PLOS ONE. Congratulations! Your manuscript is now being handed over to our production team.

Kind regards, 

on behalf of

Dr. Harapan Harapan 

Academic Editor

PLOS ONE